# A 2-hydroxybutyrate-mediated feedback loop regulates muscular fatigue

**Brennan J Wadsworth[1], Marina Leiwe[1], Eleanor A Minogue[2], Pedro P Cunha[2], Viktor Engman[3], Carolin Brombach[1], Christos Asvestis[1], Shiv K Sah-Teli[4,5], Emilia Marklund[1], Peppi Koivunen[4], Jorge L Ruas[3], Helene Rundqvist[6], Johanna T Lanner[3], Randall S Johnson[2]\***

[1]Department of Cell and Molecular Biology, Karolinska Institute, Stockholm, Sweden; [2]Department of Physiology, Development and Neuroscience, University of Cambridge, Cambridge, United Kingdom; [3]Department of Physiology and Pharmacology, Karolinska Institute, Stockholm, Sweden; [4]Faculty of Medical Biochemistry and Molecular Biology, University of Oulu, Oulu, Finland; [5]Department of Biochemistry, University of Cambridge, Cambridge, United Kingdom; [6]Department of Laboratory Medicine, Karolinska Institutet, Stockholm, Sweden

**Abstract** Several metabolites have been shown to have independent and at times unexpected biological effects outside of their metabolic pathways. These include succinate, lactate, fumarate, and 2-hydroxyglutarate. 2-Hydroxybutyrate (2HB) is a byproduct of endogenous cysteine synthesis, produced during periods of cellular stress. 2HB rises acutely after exercise; it also rises during infection and is also chronically increased in a number of metabolic disorders. We show here that 2HB inhibits branched-chain aminotransferase enzymes, which in turn triggers a SIRT4-dependent shift in the compartmental abundance of protein ADP-ribosylation. The 2HB-induced decrease in nuclear protein ADP-ribosylation leads to a C/EBPβ-mediated transcriptional response in the branched-chain amino acid degradation pathway. This response to 2HB exposure leads to an improved oxidative capacity in vitro. We found that repeated injection with 2HB can replicate the improvement to oxidative capacity that occurs following exercise training. Together, we show that 2-HB regulates fundamental aspects of skeletal muscle metabolism.

## eLife assessment

The work by Johnson and co-workers has identified an **important** role of 2-Hydroxybutyrate in skeletal muscle oxidative capacity in the early stages of exercise. Mechanistically, they show **convincing** data to support a role of 2-Hydroxybutyrate in the regulation of BCAA metabolism via SIRT4, ADP-Ribosylation, and CEBP. However, whether this is the sole mechanism and if these translate to longer exercise training regimes requires future experiments.

## Introduction

Physical exertion is a metabolic challenge that requires physiological responses throughout the body, including a dramatic change to metabolic flux patterns (*Hawley et al., 2014*). During exercise, metabolic end products accumulate and are released into the circulation; prototypical examples of such molecules include hypoxanthine and lactate. Metabolic intermediates of pathways with increased flux during exercise may also exhibit increased levels in the circulation post-exercise, particularly as an indication of pathway saturation (*Sato et al., 2022*). Accumulation of individual exercise-induced metabolites may lead to a diverse range of cellular responses in different tissues, with acute and

**\*For correspondence:**
randall.johnson@ki.se

lasting effects on physiology. Such mechanisms have been defined for succinate (**Reddy et al., 2020**), lactate (**Brooks et al., 2023**), hypoxanthine (**Yin et al., 2021**), and other exerkines (**Hawley et al., 2014**), demonstrating that the exercise-induced metabolome includes factors with significant physiological effects.

Exercise-induced lactate and succinate exert effects on physiology despite a relatively rapid return to baseline levels during recovery post-exercise. Conversely, 2-hydroxybutyrate (2HB, also known as α-hydroxybutyrate) is an exercise-induced metabolite with plasma concentrations that continue to increase for at least 3 hr after exertion (**Berton et al., 2017**; **Morville et al., 2020**; **Rundqvist et al., 2020**; **Contrepois et al., 2020**). Production of the majority of 2HB in circulation has been attributed to the liver, although 2HB accumulates in many other tissues post-exercise (**Sato et al., 2022**). 2HB is also present in disease states, as untargeted metabolomics analyses have identified 2HB as a metabolic marker for infection severity (**Bruzzone et al., 2020**), and as an indicator of metabolic disorders (**Thompson Legault et al., 2015**; **Gall et al., 2010**; **Sharma et al., 2021**). Cellular responses to accumulated 2HB have not been characterized to date, although **Sato et al., 2022**, report that exogenous administration of 2HB caused an acute shift in mouse whole body metabolism, including a reduction in the respiratory exchange ratio (RER), and increased blood glucose; indicating that 2HB has the potential to alter metabolism.

2HB is the product of a lactate dehydrogenase (LDH)-mediated reduction of 2-ketobutyrate (2 KB), which itself is a byproduct of endogenous cysteine synthesis via the transsulfuration pathway. Oxidation of 2 KB is not well characterized beyond isolated enzyme assays; these show it to be a substrate for the branched chain keto acid dehydrogenase complex (BCKDH) (**Pettit et al., 1978**). BCKDH converts 2 KB to propionyl-CoA, which can be re-carboxylated by propionyl-CoA carboxylase (PCC), producing the tri-carboxylic acid (TCA) cycle intermediate succinyl-CoA (**Mann et al., 2021**). Mammalian cells are not capable of reverse transsulfuration, that is production of methionine from cysteine; thus, the metabolic fate of 2 KB is driven by the balance between mitochondrial oxidation or reduction to 2HB. Importantly, the metabolism of exogenous 2HB has not been described. Based upon the current literature, the fate of 2HB depends on the capacity for LDH-dependent oxidation of 2HB, and thus, 2 KB metabolic homeostasis.

2HB is an alpha-hydroxy carboxylic acid similar to 2-hydroxyglutarate (2HG). The latter is a dicarboxylic acid, and an established physiological competitor of α-ketoglutarate (αKG). Both the *S*- and *R*- enantiomers of 2HG are known to competitively inhibit αKG-dependent enzymatic reactions, including the branched chain amino-transferase (BCAT) enzymes, and the much broader family of αKG-dependent dioxygenases (αKGDD) (**Intlekofer et al., 2015**). αKGDDs include regulators of the cellular responses to hypoxia, the prolyl hydroxylase (PHD) and factor inhibiting HIF (FIH), and epigenetic regulators such as lysine demethylases (KDM), among others. Regulation of these targets by endogenous or exogenous *S*-2HG is sufficient to alter numerous cell fate decisions, for example the differentiation of cytotoxic CD8 T cells (**Tyrakis et al., 2016**; **Foskolou et al., 2020**). *R*-2HG inhibition of BCAT is reported to reduce metabolism of branched chain amino acids (BCAA), leading to reduced levels of glutamate and glutathione (**McBrayer et al., 2018**). As an alpha-hydroxy acid, we hypothesised that 2HB may be a novel competitive inhibitor of αKGDDs or BCATs.

In this work, we investigate the physiological and molecular role of 2HB. We find that cells treated with 2HB demonstrate a metabolic response that leads to transcriptional regulation of BCAA degradation enzymes via a BCAT2 and SIRT4-dependent shift in compartmental protein ADP-ribosylation (ADPr). The reduced ADPr in the nuclear compartment following 2HB treatment promotes binding of the transcription factor CCAAT enhancer binding protein beta (C/EBPβ) to the promoters of BCAA degradation genes. We find that repeated administration of 2HB to mice increases oxidative capacity in exercise tests that replicates the effect of exercise training. Finally, we find repeated 2HB treatment gives rise to an improved resistance to fatigue in oxidative skeletal muscle, consistent with the improvement to oxidative metabolism. Overall, we find that 2HB induces a range of cellular responses that are consistent with it being a feedback signal from exertion, one that leads to increased capacity in the BCAA degradation pathway and improved exercise performance.

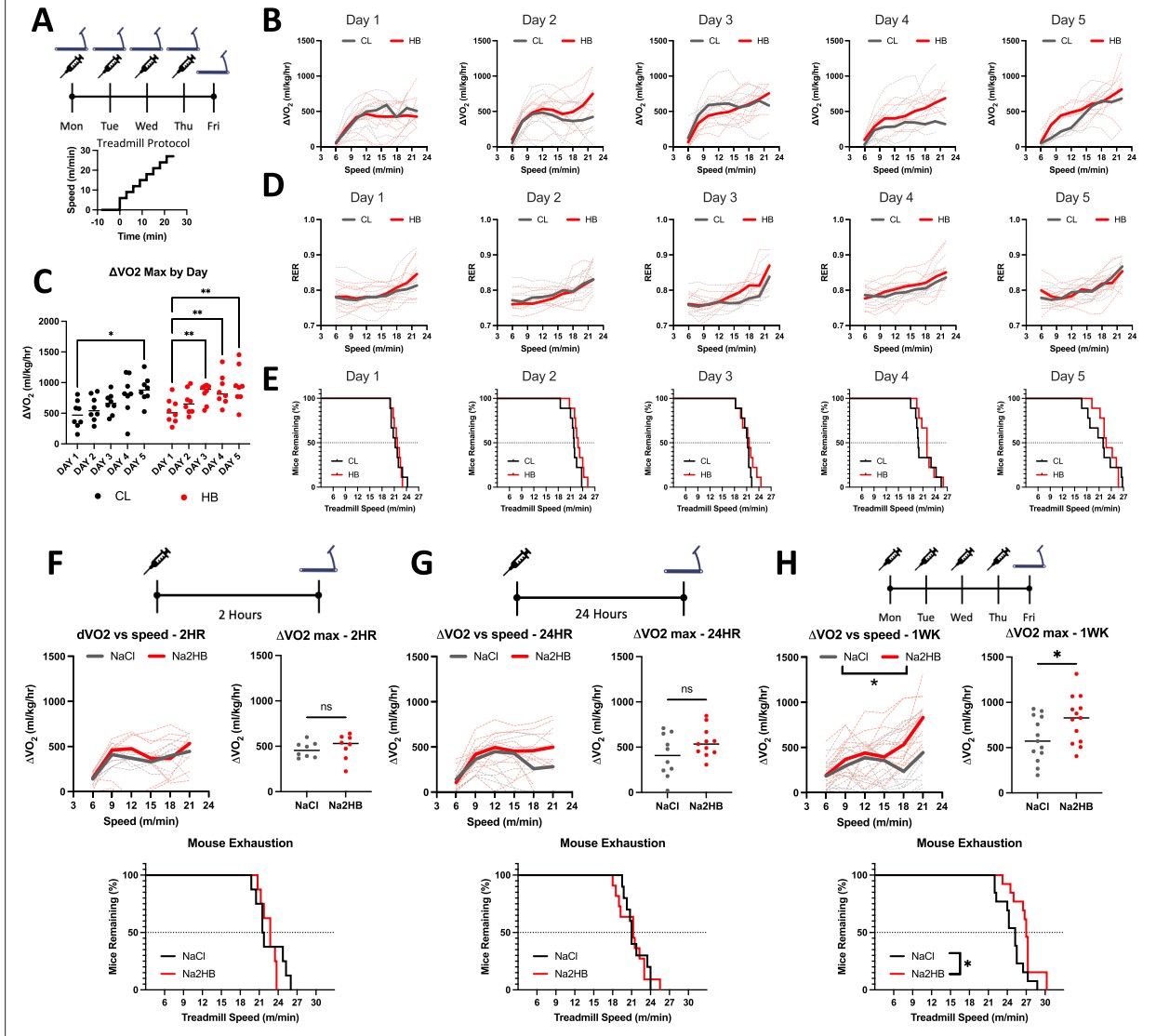

**Figure 1.** Daily 2-hydroxybutyrate treatment recapitulates the benefits of exercise training on oxidative capacity. (**A**) Mice were subjected to five days of daily incremental exercise tests to exhaustion followed immediately by injection with 1 mmol/kg of NaCl or Na2HB, N=8. (**B**) Change in VO$_2$ relative to baseline (ΔVO$_2$) versus treadmill speed. (**C**) Maximum ΔVO$_2$ compared with Day 1. Dunnett's multiple comparisons test, * $p<0.05$, ** $p<0.01$. (**D**) Respiratory exchange ratio (RER) versus treadmill speed. (**E**) Time-to-exhaustion. Mice were subject to a single incremental exercise test (**F**) 2 hr (N=8) or (**G**) 24 hr (N=11) after a single dose, or (**H**) after four daily doses (N=13) of NaCl or Na2HB. For B,D, and F-H, solid lines indicate median values and dashed lines show each replicate. Median plots clipped at 21 m/min due to mouse dropout, although mice may continue up to 30 m/min. Speed vs ΔVO$_2$ plot shows two-way ANOVA main effect * $p<0.05$, ΔVO$_2$ max comparison shows student's t-test * $p<0.05$. Time-to-exhaustion plots show Log-Rank test, * $p<0.05$.

The online version of this article includes the following source data and figure supplement(s) for figure 1:

**Source data 1.** Raw data for *Figure 1*.

**Figure supplement 1.** Supplemental data regarding exercise experiments.

**Figure supplement 1—source data 1.** Raw data for *Figure 1—figure supplement 1*.

# Results
## Daily 2HB treatment accelerates the metabolic response to exercise in mice

As described above, physical exertion induces a persistent increase to circulating 2HB. To investigate the role of 2HB in whole animal metabolism, we first subjected mice to daily incremental exhaustive

exercise tests for five consecutive days. Immediately following each test, mice were injected with 1 mmol/kg of Na2HB, or an equivalent dose of NaCl as a control (*Figure 1A*). Mouse basal $VO_2$, maximum $\Delta VO_2$ (change in $VO_2$ relative to basal $VO_2$), RER, and $\Delta VO_2$ throughout exercise were similar across treatment groups on the first day, prior to group randomization (*Figure 1B–D*, *Figure 1— figure supplement 1A*). Median mouse $\Delta VO_2$max was 507 mL/kg/hr on the first day (*Figure 1C*) and mice tended to reach exhaustion during the 21 m/min phase (*Figure 1E*). Beyond 18 m/min, mice either further increased their $VO_2$ or reached exhaustion. This is demonstrated by the lack of correlation between $\Delta VO_2$ during 18 m/min and time-to-exhaustion, but a significant correlation between either $\Delta VO_2$ during 21 m/min or maximum $\Delta VO_2$ and time-to-exhaustion (*Figure 1—figure supplement 1B*). These data demonstrate that increased oxidative capacity predicts improved performance in incremental exercise tests.

Mouse $\Delta VO_2$max tended to increase with subsequent exercise tests and had significantly improved to a median of 877 mL/kg/hr in the control group by day 5 (*Figure 1C*). Accordingly, progressively more mice reached the 24 m/min phase before exhaustion (*Figure 1E*). Mice treated with 2HB showed improvement to $\Delta VO_2$max in subsequent exercise tests as early as day 3, and increased $\Delta VO_2$max up to 930 mL/kg/hr on day 5 (*Figure 1C*). These data demonstrate a benefit to exercise performance from a brief one-week protocol of exercise training in mice and suggest that combination with exogenous 2HB may accelerate this effect.

## Daily 2HB treatment recapitulates the benefits of exercise training on oxidative capacity

We next investigated the effects of exogenous 2HB alone. Firstly, we assessed the effects of acute administration of 1 mmol/kg of Na2HB. For mice at rest, we observed no effect of exogenous 2HB on oxygen consumption or RER compared with NaCl control over the first 60 min, consistent with previous reports for this dose (*Figure 1—figure supplement 1C and D*; *Sato et al., 2022*). We next administered 2HB 2 hr prior to an exhaustive exercise test. Both 2HB-treated and control mice reached a $\Delta VO_2$max of approximately 500 mL/kg/hr, with no difference in RER, time-to-exhaustion, resting $VO_2$, or $\Delta VO_2$ during the exercise tests (*Figure 1F*, *Figure 1—figure supplement 1E and F*). These data indicate that exogenous 2HB at a dose of 1 mmol/kg does not acutely alter resting metabolism or oxygen consumption during exercise.

Next, we investigated whether treatment with 2HB would alter exercise performance on subsequent days. First, a cohort of mice was treated with 2HB 24 hr prior to an exhaustive exercise test. In these mice there was again no difference between groups in terms of resting $VO_2$, $\Delta VO_2$ or RER during the exercise test, time-to-exhaustion, or $\Delta VO_2$max (*Figure 1G*, *Figure 1—figure supplement 1E and F*). When mice were treated with 2HB each day for four days with the exhaustive exercise test on the fifth day, 2HB-treated mice exhibited an increased $\Delta VO2$ during the exercise test, with differences more apparent at greater treadmill speeds (*Figure 1H*). Oxygen consumption at higher treadmill speeds correlated with greater time-to-exhaustion, indicating effective oxidative capacity as opposed to metabolic inefficiency (*Figure 1—figure supplement 1G*). Mice treated with 2HB exhibited a greater $\Delta VO_2$max and greater time-to-exhaustion, but had similar basal $VO_2$ and RER levels throughout exercise, relative to exercising NaCl-treated controls (*Figure 1H*, *Figure 1—figure supplement 1E and F*). Control mice exhibited a median $\Delta VO_2$max of 572 ml/kg/hr. However, 2HB-treated mice exhibited a median $\Delta VO_2$max of 828 mL/kg/hr, similar to exercise-trained mice (*Figure 1C and H*). These data indicate that repeated 2HB treatment replicates the improvement to mouse oxidative capacity and exercise performance produced from daily exercise training.

## 2HB levels are correlated with metabolic indicators of BCAA degradation in murine datasets

To determine which metabolic pathways 2HB may affect, we were next interested in determining which metabolites are most correlated with 2HB post-exercise. We analyzed two published datasets reporting untargeted metabolomics in mice subjected to a treadmill-based exercise protocol, or sham control (*Sato et al., 2022*; *Rundqvist et al., 2020*). Both studies observe a strong increase to 2HB in the circulation and in all assessed organs post-exercise (*Figure 2*, *Figure 2—figure supplement 1A*). A correlation analysis demonstrates a reliable positive correlation between circulating 2HB and markers of BCAA metabolism (*Figure 2A*). Overrepresentation analysis of BCAA-related metabolites

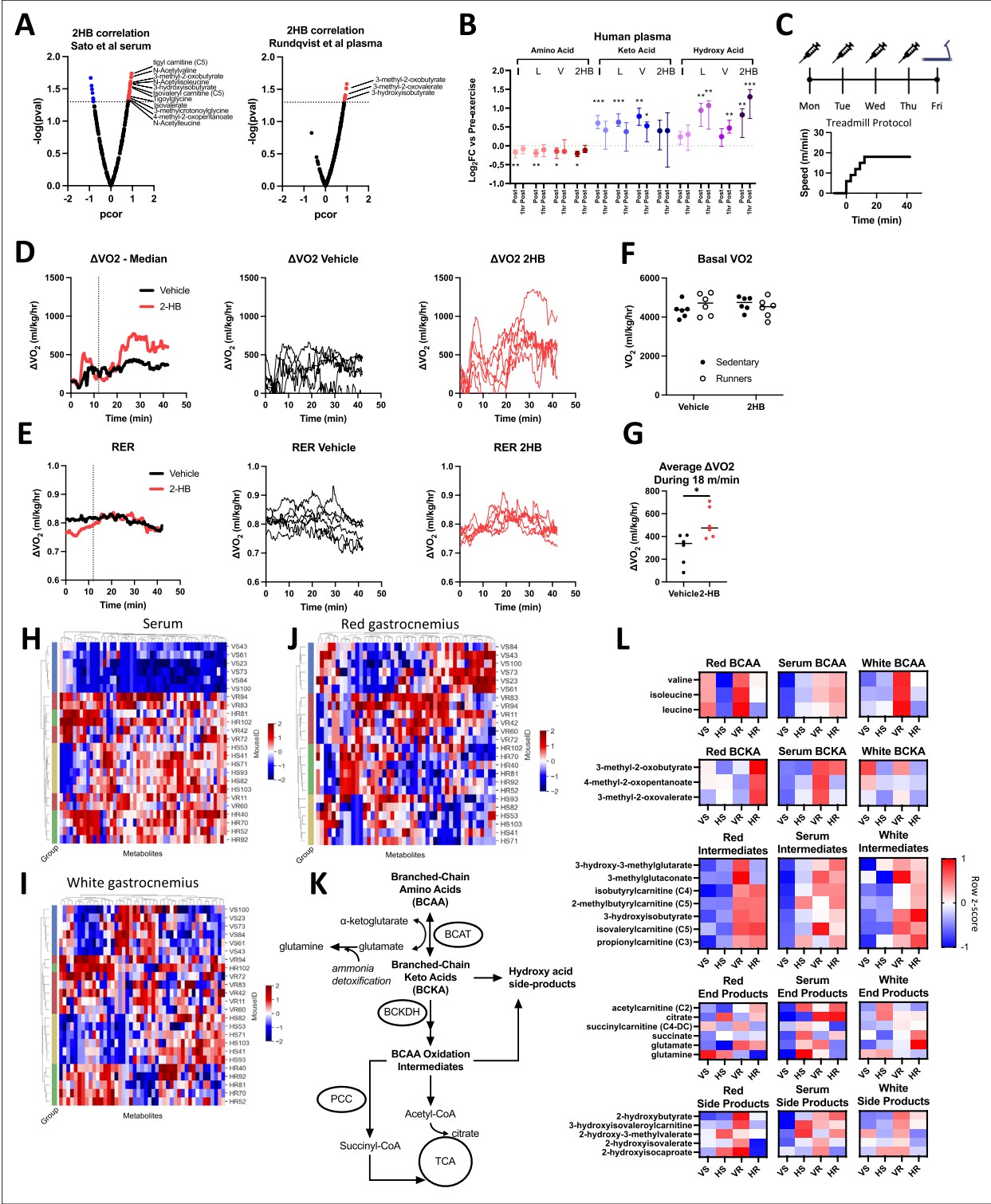

**Figure 2.** Exercise datasets suggest relation between 2-hydroxybutyrate and BCAA metabolism. (**A**) Volcano plots display partial correlation coefficients for metabolites present in the blood that correlate with 2HB levels in mice that have undergone exercise. BCAA degradation metabolites indicated. Data in A are from ***Sato et al., 2022*** (*left*) and ***Rundqvist et al., 2020*** (*right*). (**B**) Levels of amino acid, keto-acid, and hydroxy acid versions of the BCAA and 2HB are displayed for human subjects comparing pre-exercise measures with those immediately post exercise and 1-hr post-exercise. Data from ***Rundqvist et al., 2020***, N=8, median ±interquartile range. Dunnett's multiple comparisons test made against pre-exercise levels, *** p<0.001, ** p<0.01, * p<0.05. (**C**) Mice were treated with 1 mmol/kg of NaCl or Na2HB for four days prior to a single endurance exercise protocol consisting of a gradual warm-up and 30 min of running at 18 m/min. Sham control mice placed on motionless treadmill. (**D**) ΔVO₂ and (**E**) RER plots, Left-to-Right:

*Figure 2 continued*

group medians, vehicle-treated, and 2HB-treated. (**F**) No effect of 2HB on basal $VO_2$, Dunnett's multiple comparisons test. (**G**) 2HB-treated mice show increased average $\Delta VO_2$. Student's t-test, * p<0.05. Untargeted metabolomics was conducted on serum, red gastrocnemius, and white gastrocnemius from mice in figures **C-G**. Heat maps of top 50 most differentially abundant metabolites in (**H**) serum, (**I**) white gastrocnemius, and (**J**) red gastrocnemius. Unique mouse ID displayed with group abbreviations: 'VS' vehicle-sedentary, 'HS' 2HB-sedentary, 'VR' vehicle-runner, 'HR' 2HB-runner. (**K**) BCAA degradation pathway. (**L**) Heat map displays metabolites in the BCAA degradation pathway. For D-L, N=6. Heat maps display standardized row medians.

The online version of this article includes the following source data and figure supplement(s) for figure 2:

**Source data 1.** Raw data for *Figure 2*.

**Figure supplement 1.** Supplemental data regarding 2-hydroxybutyrate and BCAA metabolomics.

**Figure supplement 1—source data 1.** Raw data for *Figure 2—figure supplement 1*.

positively correlated with 2HB post-exercise yielded p values of $3.8\times10^{-5}$ and $1.6\times10^{-2}$ for the two studies. These data suggest that accumulation of 2HB is correlated to the metabolic flux through the BCAA degradation pathway.

## Human exercise studies demonstrate the persistence of serum 2HB post-exercise

We further investigated this pattern by assessing the plasma metabolomics of human subjects in a previously published dataset from our group. In this study, blood was sampled prior to exercise, immediately post-exercise, and then 1-hr post-exercise. Few markers of BCAA degradation are available in the human dataset, so we focused on the amino acid, the keto acid, and the hydroxy acid versions of leucine, isoleucine, valine, and 2HB. Transamination of BCAA to the respective keto acid is the first step towards mitochondrial oxidation. In line with this, BCAA levels are reduced immediately post-exercise compared with pre-exercise levels, while keto acids and hydroxy acids are increased (*Figure 2B*). The accumulation of 2HB alongside increased BCAA degradation is consistent with a model wherein BCAAs compete with 2 KB for early oxidation steps, for example the BCKDH and PCC-mediated reactions.

Indicating a return to homeostasis, 1-hr after completion of exercise, BCAAs increase and keto acids decrease towards pre-exercise levels (*Figure 2B*). The hydroxy acids are intriguing, as these remain significantly greater than pre-exercise levels even 1-hr after completion of exercise, suggesting slow clearance of all members of this metabolite group, including 2HB. Among the hydroxy acids, 2HB is the standout, with the greatest fold-increase at the 1-hr time point and the greatest absolute abundance in both human and mouse datasets (*Rundqvist et al., 2020*). Collectively, these data describe a positive correlation for the production of 2HB in proportion with production of branched chain keto acids (BCKA) during exercise. The slow clearance of 2HB and the branched chain hydroxy acids (BCHA) represent a possible indication of saturation in the capacity for the oxidation of 2 KB and BCKAs.

## Repeated 2HB treatment alters BCAA metabolite balance following exercise

We next investigated whether repeated treatment of 2HB for four days would alter BCAA metabolism during an exercise test on the fifth day. To limit mouse dropout from exhaustion, the exercise protocol increased speed only up to 18 m/min. This speed was held for 30 min for all mice (*Figure 2C*). There was no effect of 2HB treatment on basal $VO_2$, or on RER throughout the exercise protocol (*Figure 2D–F*). 2HB-treated mice tended to spend more time running on the treadmill, indicated by an increased $\Delta VO_2$ during the 18 m/min segment of the exercise protocol (*Figure 2D and G*). These data demonstrate again that repeated 2HB treatment increases oxidative capacity during exercise.

We hypothesized that the change to exercise performance, in both the incremental exercise tests to exhaustion and moderate exercise tests, was in part due to a change in skeletal muscle metabolism during exercise. Untargeted metabolomics of mouse serum, red gastrocnemius, and white gastrocnemius collected immediately post-exercise shows modest clustering of mice based on treatment group in the serum, separating vehicle-treated sedentary controls from other treatment groups (*Figure 2H*). Conversely, we observe strong clustering in the red and white gastrocnemius, delineating

each experimental group (*Figure 2I and J*). These data begin to suggest that skeletal muscle is a site of metabolic regulation by 2HB.

When we visualize the data trends in the BCAA metabolic pathway within each tissue (*Figure 2K and L*), a shift in the BCAA, BCKA, and BCHA appears to be exercise and 2HB dependent; specifically 2HB treatment appears to reduce BCAA levels in red gastrocnemius at rest and during exercise, but increase BCKA and decrease the hydroxy acids during exercise (*Figure 2L*). The levels of intermediates within the BCAA degradation pathway increase in each compartment with exercise, with no strong effect of 2HB. These data suggest a shift in red gastrocnemius BCAA homeostasis induced by 2HB treatment.

To compare the effects of 2HB given the quantifiable difference in exercise performance (*Figure 2D–G*), we adjusted metabolite values based on VO$_2$ measures during the exercise protocol or sham control. This adjustment yields values that indicate whether the levels of a given metabolite were high or low relative to the matching VO$_2$ measures for each mouse. We observed a number of differences and trends in the red gastrocnemius muscle: reduced levels of BCAA at rest and post-exercise, increased BCKA post-exercise, reduced BCHA post-exercise, and reduced 2HB post-exercise (*Figure 2—figure supplement 1B*). In white gastrocnemius we only observe a reduction to BCAA and the BCHA post-exercise (*Figure 2—figure supplement 1C*). Each of these effects are weaker in the serum VO$_2$-adjusted data (*Figure 2—figure supplement 1D*). These data demonstrate that 2HB treatment alters muscle BCAA metabolic balance in excess of the changes to mouse VO$_2$ during exercise.

## 2KB is a fuel for oxidative metabolism

The metabolomics analysis led us to hypothesize that physiological 2HB accumulation is associated with competition for the BCAA degradation pathway, limiting oxidation of 2 KB. However, despite validation as a substrate for BCKDH in isolated enzyme assays (*Pettit et al., 1978*), 2 KB has not been validated as a potential fuel for mitochondrial oxidative metabolism. Thus, we next aimed to investigate the potential of 2HB and 2 KB as fuel for oxidative metabolism in vitro and dependency of 2 KB oxidation on the BCAA degradation pathway. We subjected cells to Seahorse assays monitoring the oxygen consumption rate (OCR) during a standard 'mitochondrial stress test' protocol following acute administration of 500–5000 μM of either Na2HB, Na2KB, or NaCl control (*Figure 3A and B*). With C2C12 myoblasts, we found a modest increase to OCR immediately following 2 KB addition, with no response to 2HB (*Figure 3A and B*). Presence of 2HB or 2 KB provided a small increase to the maximum OCR. We hypothesized that the standard assay media saturated cells with fuel, leaving little capacity for 2HB or 2 KB as a fuel. We therefore repeated the assays with no pyruvate in the medium. Similarly, an acute uptick in OCR was observed upon addition of 2 KB, but not 2HB (*Figure 3A and B*). There was now a dose-dependent increase to maximal OCR in cells treated with 2 KB, up to a similar magnitude when pyruvate was included. Addition of 2HB was not able to recapitulate the maximum OCR observed when pyruvate was present in the media. These trends could be replicated with mouse embryonic fibroblasts (MEF), although with a lower magnitude (*Figure 3—figure supplement 1A*). These data demonstrate that 2 KB is a substantially greater fuel for oxidative metabolism than 2HB.

It is possible that the increase to maximal OCR induced by 2 KB is due to 2 KB promoting glycolysis (*Sullivan et al., 2015*; *Lesner et al., 2020*), instead of 2 KB being used as a fuel. To investigate the importance of the BCAA degradation pathway for 2 KB metabolism, we repeated these experiments in cells transfected with siRNA against *BCKDHA*, *PCCA*, and *HPRT* as an off-target control, or non-targeting siRNA control (*Figure 3—figure supplement 1B*). Transfection alone had insignificant effects on maximum OCR, as demonstrated by cells treated with NaCl. Addition of 500 μM 2 KB increased maximum OCR as before, however, this was inhibited in cells transfected with siRNA against *PCCA* (*Figure 3C*). These data suggest that an intact BCAA degradation pathway is required for 2 KB promotion of maximal OCR in vitro, indicating that 2 KB can be used as a fuel for oxidative metabolism.

## 2HB is slowly metabolized by LDH

The poor suitability of 2HB as a fuel relative to 2 KB suggests slow conversion to 2 KB. Interconversion between 2HB and 2 KB is proposed to occur via LDH. Thus, we aimed to quantify the relative rate of 2HB oxidation and 2 KB reduction. We conducted activity assays using LDH isolated from bovine skeletal muscle or bovine heart. The former is enriched for LDH isoenzymes with M subunits from *LDHA*

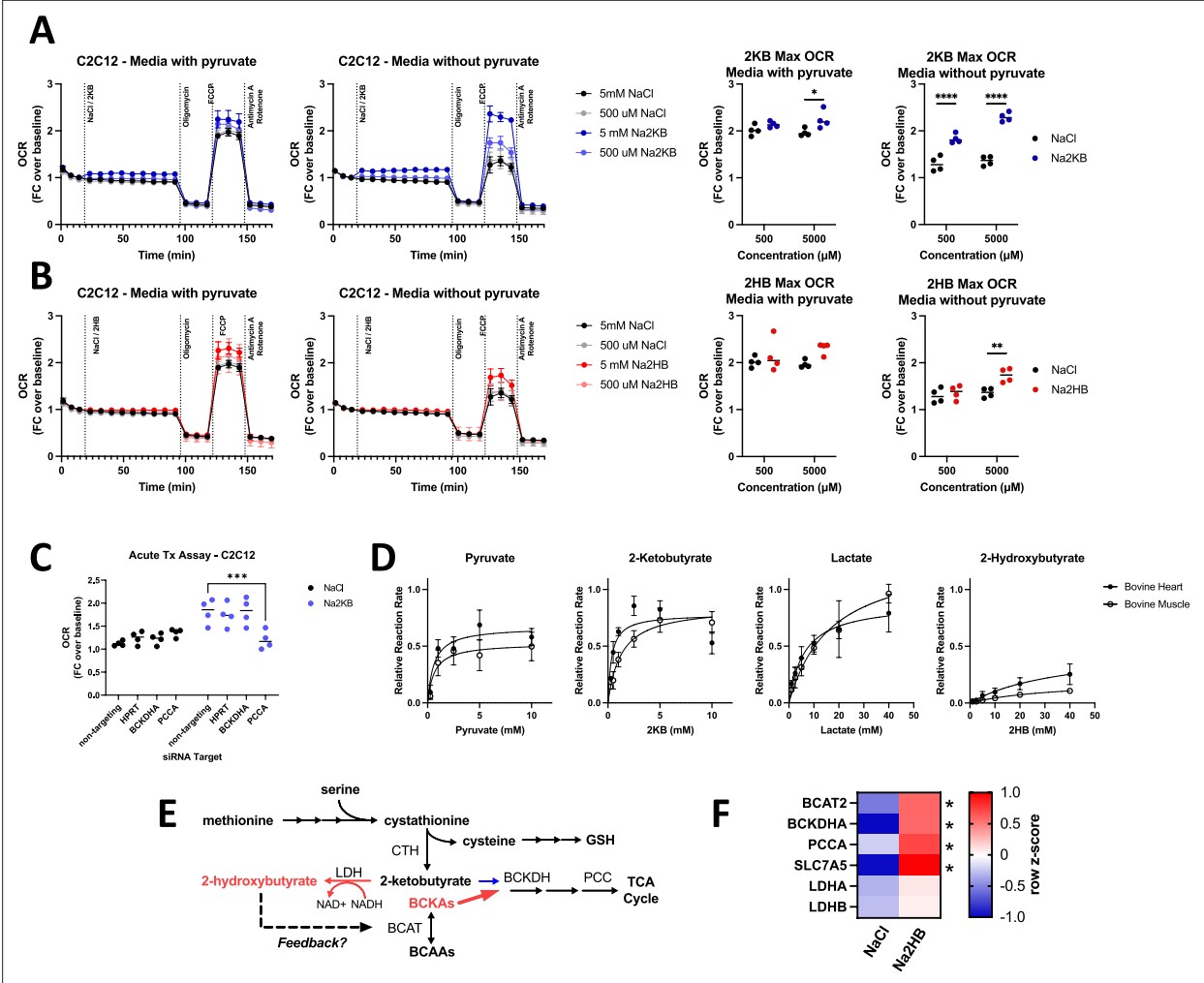

**Figure 3.** 2-Ketobutyrate oxidation depends upon an in tact BCAA degradation pathway. (**A,B**) Oxygen consumption rate (OCR) during mitochondrial stress test with acute administration of (**A**) Na2KB or (**B**) Na2HB as indicated. N=4 independent assays, mean ± SD. Basal and maximum OCR values displayed with Sidak's multiple comparisons test displayed, * p<0.05, **** p<0.0001. (**C**) Maximum OCR values from mitochondrial stress tests in C2C12 transfected with siRNA against indicated targets, N=4 independent assays mean ± SD. Sidak's multiple comparisons test, *** p<0.001. (**D**) Relative reaction rate of LDH assays using indicated substrate. Data are normalized to the maximum reaction rate for a given experiment day for each reaction direction. Data are a summary of five independent experiment days, mean ± SD. (**E**) Summary figure; we propose that accumulation of 2HB, as occurs post-exertion, is an indicator of saturation in the BCAA degradation pathway, leading to reduction of 2 KB instead of mitochondrial oxidation and the potential for metabolic feedback via 2HB. (**F**) Heatmap summarizing RT-qPCR for C2C12 myoblasts treated with 500 µM of Na2HB or NaCl for 8 hr. N=4 independent assays, median row z-score displayed with Sidak's multiple comparisons test, * p<0.05.

The online version of this article includes the following source data and figure supplement(s) for figure 3:

**Source data 1.** Raw data for *Figure 3*.

**Figure supplement 1.** Supplemental data regarding the BCAA degradation pathway.

**Figure supplement 1—source data 1.** Raw data for *Figure 3—figure supplement 1*.

expression, while the latter is enriched for LDH isoenzymes with H subunits due to predominant *LDHB* expression. As previously reported, 2 KB was an efficient substrate for both LDH enzymes, with a maximum relative reaction rate greater than pyruvate (*Figure 3D*; *von Morze et al., 2016*). However, 2HB exhibited a maximum reaction rate that was only ~30% that of lactate using bovine heart LDH, and ~15% of lactate using bovine skeletal muscle LDH (*Figure 3D*). These data support a mechanism for the efficient production of 2HB from 2 KB, leading to long-lasting 2HB accumulation post-exercise.

Further, we found very low reaction rates for reduction of the BCKA, with a rate between ~2 and 10% the reaction rate of pyruvate. The greatest reaction rate observed was for the keto acid of valine,

2-oxo-3-methylbutyrate, using bovine heart LDH (*Figure 3—figure supplement 1C*). No signal was detected above background for the oxidation reaction of any BCHA. These data would be consistent with the relatively low abundance of BCHA, and their slow rate of clearance.

## 2HB metabolic feedback occurs via BCAT inhibition

The slow metabolic processing of 2HB makes it likely to accumulate in cells and trigger a functional response. Based on the effects of 2HB treatment on BCAA, BCKA, and BCHA levels post-exercise (*Figure 2*), we hypothesized that 2HB provides metabolic feedback to the BCAA degradation pathway (*Figure 3E*). As an initial test, we found that culturing C2C12 myoblasts with 500 µM of 2HB increased gene expression of multiple factors involved in BCAA degradation, including those involved in 2 KB oxidation such as *BCKDHA*, and *PCCA* (*Figure 3F*).

Based on the structural similarities between 2HB and 2HG, we hypothesized that 2HB may act as a competitive inhibitor of BCAT enzymes. To test this possibility, we conducted activity assays of recombinant human BCAT2 (rhBCAT2) using leucine as a constant substrate and αKG as a varied concentration substrate. In these assays, we find a dose dependent reduction to maximal rhBCAT2 activity with increasing concentrations of 2HB in the reaction well (*Figure 4A*). We conducted similar assays using rhGOT1, an aspartate-oxaloacetate aminotransferase, but found no effect of 2HB on the reaction rate (*Figure 4—figure supplement 1A*). Given the modification of BCAT activity, it seemed plausible that 2HB may compete with αKG for active sites in αKGDDs. However, in assays of isolated HIFP4H1 and HIFP4H2, no more than 5% inhibition was observed for concentrations of 2HB up to 10 mM, and similarly little inhibition was observed for assays of isolated KDM6A (*Figure 4—figure supplement 1 Supplementary file 1*). These data indicate that 2HB inhibits specifically BCAT activity.

Inhibition of BCAT enzymes may induce a range of metabolic disruptions. To broadly investigate cellular response to acute exogenous 2HB, we quantified cellular NAD+/NADH ratio after addition of 500 µM 2HB to culture media. In C2C12 lysates harvested 60 min following addition of 2HB we found an increase to the NAD+/NADH ratio, indicating a drop in the cellular reducing potential (*Figure 4B*). If cells were harvested after overnight culture, NAD+/NADH ratio had rebounded. There was no effect of treating cells with 2 KB, indicating that this metabolic response was specific to 2HB. These findings were replicated in MEFs (*Figure 4B*). To determine if the NAD+/NADH shift induced by 2HB exposure was dependent upon BCAT expression, we transfected C2C12 cells with a vector driving constitutive expression of DDK-tagged hBCAT2, or empty vector control. The increase to NAD+/NADH could be replicated in vector control cells, but the NAD+/NADH ratio in hBCAT2-transfected cells was insensitive to 2HB addition (*Figure 4C*, *Figure 4—figure supplement 1B*). These data describe an acute metabolic shock induced by addition of 2HB to cells, dependent upon BCAT inhibition.

## 2HB treatment stimulates SIRT4 ADP ribosyltransferase activity

Increases to cellular NAD +are sensed by Sirtuins, a family of deacetylase enzymes that modify cellular metabolism. Sirtuin 4 (SIRT4) is localised to the mitochondria and is previously described to induce increased expression and post-translational activation of the BCAA degradation pathway (*Anderson et al., 2017*; *Zaganjor et al., 2021*). We hypothesized that SIRT4 would be stimulated by the acute shift in NAD +induced by exogenous 2HB, and that SIRT4 may drive the cellular response to 2HB. First, we transfected MEFs and C2C12 myoblasts with siRNA against SIRT4 and checked for transcriptional induction of BCAA degradation pathway genes following overnight culture in media supplemented with 500 µM 2HB or NaCl control (*Figure 4—figure supplement 1C* C). We found induction of genes including *BCAT2*, *BCKDHA*, *BCKDHB*, *PCCA*, *SLC7A5*, as well as *LDHB* and *LDHA* in both MEFs and C2C12s treated with 2HB compared with controls. Transfection with siRNA against SIRT4 disrupted the induction of these genes when cells were treated with 2HB (*Figure 4D*). These data indicate that expression of SIRT4 is critical for the transcriptional feedback induced by 2HB treatment.

SIRT4 is perhaps better characterized as an ADP ribosyl transferase than as a deacetylase (*Min et al., 2018*). Addition of ADPr to protein requires consumption of local NAD+. Replacing the consumed NAD +can cause a compartmental shift in the supply of NAD+, and therefore the source of protein ADPr (*Hopp et al., 2021*). Thus, metabolic activity in one compartment may induce transcriptional change via regulating ADPr of transcription factors within the nucleus. To investigate the ability of SIRT4 to regulate compartmental ADPr, we separated cytoplasmic and nuclear extracts of C2C12 myoblasts transfected with siRNA against *SIRT4*, or non-targeting control, treated with 500 µM 2HB or

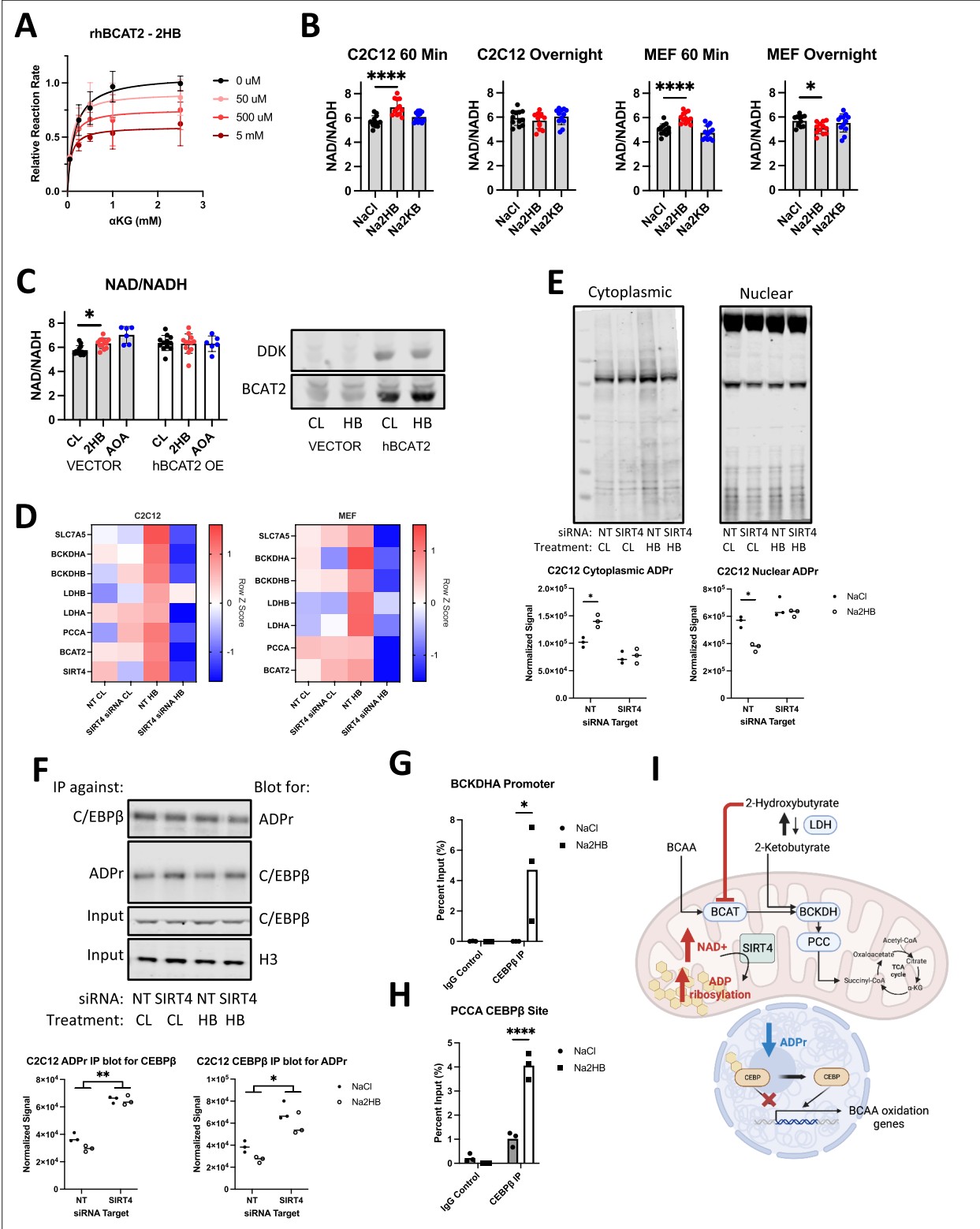

**Figure 4.** 2-Hydroxybutyrate feedback mecanism. (**A**) Relative reaction rate for recombinant human BCAT2 activity assays. Leu was kept at a constant concentration of 5 mM, 2HB concentrations are indicated in the legend. Data are normalized to the maximum reaction rate for a given experiment day and data represent three independent experiment days, mean ± SD. (**B**) Indicated metabolites were added to culture media for indicated time prior to cell harvest for NAD +and NADH assays, NAD+/NADH ratio displayed. N=8 independent assays, Dunnett's multiple comparisons test, **** p<0.0001, * p<0.05. (**C**) C2C12 myoblasts were transfected with vector control or DDK-tagged hBCAT2 before repeating the 60-min time point NAD +and NADH

*Figure 4 continued on next page*

*Figure 4 continued*

assays. 1 mM Aminooxyacetic acid (AOA) included as positive control. N=12 independent assays, Dunnett's multiple comparisons test, * p<0.05. (**D**) RT-qPCR gene expression for factors related to 2HB and BCAA metabolism in MEFs and C2C12 transfected with siRNA against SIRT4 or non-targeting control, and treated with 500 μM NaCl or Na2HB for 8 hr. N=4 independent assays, median row z-score displayed. (**E**) Total protein ADP ribosylation western blots from cytoplasmic and nuclear extracts of C2C12 cells transfected with siRNA against SIRT4 or non-targeting control and treated with 500 μM NaCl or Na2HB for 16 hr. Representative blots shown and quantification from three independent experiments. N=3, Sidak multiple comparisons test, * p<0.05. (**F**) Immunoprecipitation was conducted on C2C12 nuclear extracts using either antibody against C/EBPβ or poly/mono-ADP ribose, with resultant membranes probed using the opposite antibody. Total C/EBPβ and histone 3 were detected in input samples. N=3, two-way ANOVA main effect, ** p<0.01, * p<0.05. (**G**) ChIP-qPCR conducted on C2C12 myoblast lysates treated with 500 μM of NaCl or Na2HB for 8 hr. (**I**) Summary figure displaying proposed mechanism.

The online version of this article includes the following source data and figure supplement(s) for figure 4:

**Source data 1.** Raw data for *Figure 4* and western blots.

**Source data 2.** Labeled western blots.

**Source data 3.** Unlabeled western blots.

**Figure supplement 1.** Supplemental data regarding the 2-hydroxybutyrate feedback mechanism.

**Figure supplement 1—source data 1.** Raw data for *Figure 4—figure supplement 1*.

**Figure supplement 1—source data 2.** Labeled western blots.

**Figure supplement 1—source data 3.** Unlabeled western blots.

NaCl control for 16 hr. In western blots detecting total poly/mono-ADPr, we see that 2HB treatment increases ADPr in the cytoplasmic compartment, while reducing ADPr in the nuclear extracts, each dependent upon SIRT4 (*Figure 4E*, *Figure 4—figure supplement 1D*). We were next interested in whether 2HB-induced SIRT4 activity might alter the ADPr of a transcription factor relevant to regulation of metabolism. C/EBPβ has been described to regulate adipogenesis (*Ryu et al., 2018*), which is also known to involve SIRT4-dependent regulation of BCAA metabolism (*Zaganjor et al., 2021*; *Ryu et al., 2018*). Importantly, C/EBPβ transcriptional activity was shown to be dependent upon compartmental ADPr, wherein C/EBPβ ADPr reduces its transcriptional activity (*Ryu et al., 2018*). Therefore, we conducted immunoprecipitation experiments pulling down total poly/mono-ADPr protein from C2C12 nuclear extracts and blotting for C/EBPβ, or vice versa. We observed a strong effect of SIRT4 knockdown leading to increased ADPr of C/EBPβ, and a trend towards reduced C/EBPβ ADPr in cells treated with 500 μM 2HB for 8 hr (*Figure 4F*). It is possible that a small shift in ADPr may lead to significant changes to C/EBPβ transcriptional activity. We therefore continued by conducting chromatin immunoprecipitation experiments pulling down C/EBPβ and conducted qPCR for different promoter regions in the genes of *BCKDHA* and *PCCA* containing C/EBPβ consensus binding sites. We found that treating cells with 500 μM 2HB for 8 hr stimulated binding of C/EBPβ to both gene promoters (*Figure 4G and H*). In summary, we find that treating cells with exogenous 2HB induces a metabolic response via BCAT inhibition, which stimulates SIRT4 to trigger a compartmental change in protein ADPr, leading to C/EBPβ binding to *BCKDHA* and *PCCA* genes (*Figure 4I*). These data present a mechanism for feedback on the BCAA degradation pathway induced by 2HB accumulation.

## 2HB promotes oxidative metabolism in vitro via the BCAA degradation pathway

We hypothesized that overnight culture with 2HB would increase the expression of BCAA degradation pathway genes and thus promote oxidative metabolism. MEFs and C2C12s were treated overnight with 500 μM of Na2HB, or NaCl control prior to running a mitochondrial stress test or glycolysis stress test to quantify OCR and extracellular acidification rate (ECAR) respectively. Overnight treatment with either 2HB increased both the basal and maximal OCR in MEFs and C2C12s (*Figure 5A and B*). MEFs exhibited an increased ECAR, indicative of increased glycolytic capacity from treatment with 2HB (*Figure 5—figure supplement 1A*). Collectively, these data indicate increased metabolic activity, and particularly an increase to oxidative capacity from overnight culture with 2HB.

We hypothesized that the increased oxidative capacity induced by 2HB was dependent upon SIRT4 stimulation and activity of the BCAA degradation pathway. To address this possibility, we transfected C2C12s and MEFs with siRNA against SIRT4, BCKDHA, or non-targeting control. Cells were then cultured overnight in media supplemented with 500 μM Na2HB or NaCl, then subjected to a

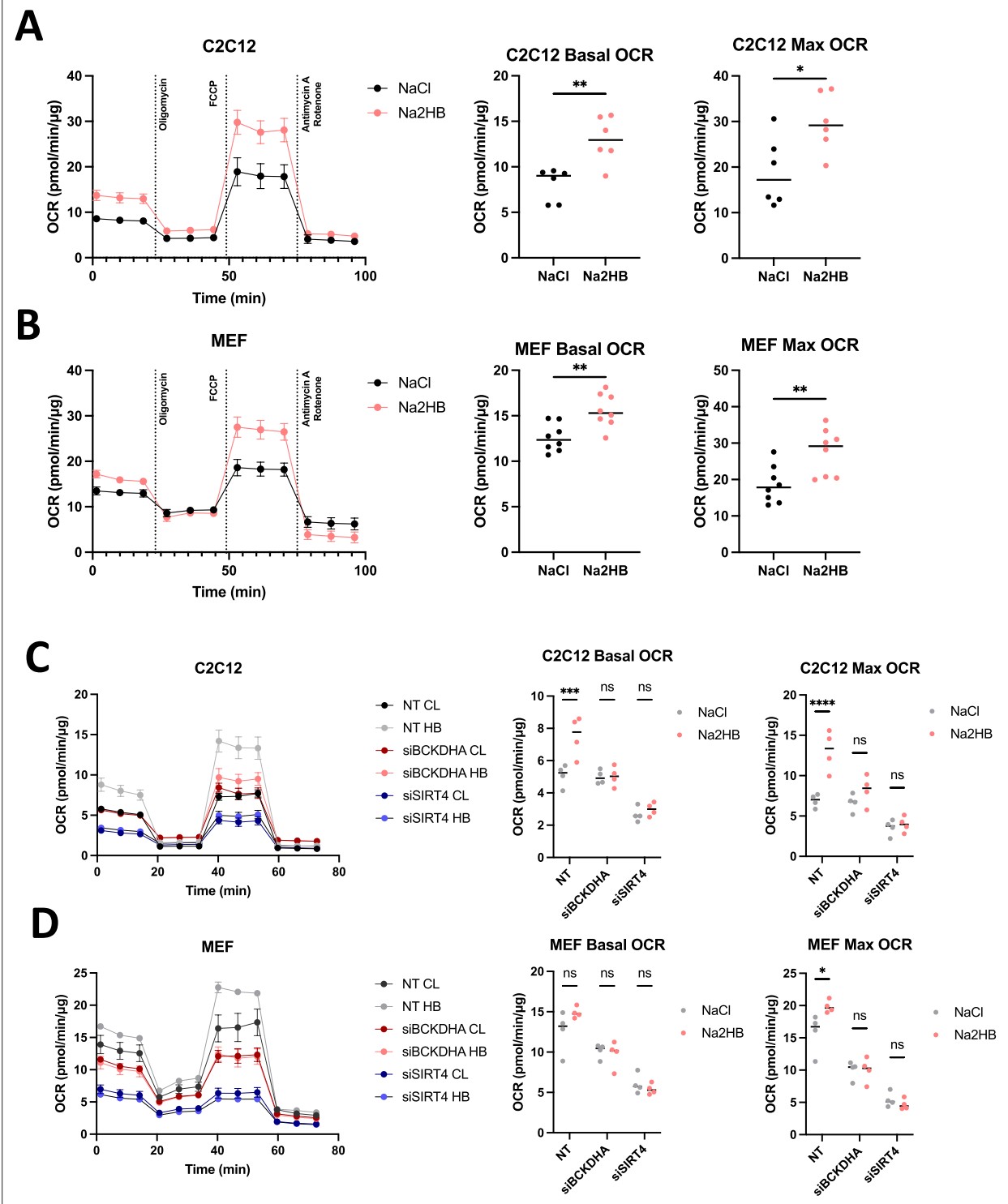

**Figure 5.** 2-Hydroxybutyrate promotes oxidative metabolism in vitro. (**A**) C2C12 and (**B**) MEFs cultured overnight in media containing 500 µM of Na2HB, or NaCl as indicated, prior to mitochondrial stress test, mean ± SD. Basal and maximum OCR values displayed with Student's t-test results; N=6 independent assays ** p<0.01, * p<0.05. OCR during mitochondrial stress tests, basal OCR, and maximum OCR for (**C**) C2C12s and (**D**) MEFs transfected with siRNA against indicated gene targets. N=4 independent assays, mean ± SEM Sidak's multiple comparisons test, **** p<0.0001, *** p<0.001, * p<0.05.

The online version of this article includes the following source data and figure supplement(s) for figure 5:

**Source data 1.** Raw data for *Figure 5*.

*Figure 5 continued on next page*

*Figure 5 continued*

**Figure supplement 1.** Supplemental data regarding the effects of 2-hydroxybutyrate treatment in vitro.

**Figure supplement 1—source data 1.** Raw data for *Figure 5—figure supplement 1*.

**Figure supplement 1—source data 2.** Labeled western blots.

**Figure supplement 1—source data 3.** Unlabeled western blots.

mitochondrial stress test. As before, both MEFs and C2C12s treated with 2HB exhibited an increased basal and maximal OCR (*Figure 5C and D*). Transfection with siRNA against SIRT4 reduced both basal and maximal OCR and abolished the effect of 2HB (*Figure 5C and D*). Transfection with siRNA against BCKDHA induced a less drastic reduction to basal and maximal OCR, but also showed no response to 2HB treatment (*Figure 5C and D*). These data show that SIRT4 and the BCAA degradation pathway, including the early step catalyzed by BCKDHA, is critical to the increased oxidative capacity induced by 2HB treatment.

## 2HB-induced increase to oxidative capacity is maintained with chronic treatment

We were interested in whether the increase to cellular OCR would be maintained with chronic treatment. MEFs and C2C12s were treated with fresh media containing 500 µM of Na2HB or NaCl control daily for three days and assays run on the fourth day. We found chronic treatment with 2HB to result in increased maximum OCR in seahorse assays, to a similar effect size as overnight treatment (*Figure 5—figure supplement 1B*). We were interested if other mechanisms could be at play with chronic 2HB treatment. 2HB has been reported to promote cervical tumour cell survival via stimulation of the methyltransferase DOT1L (*Liu et al., 2018*). We therefore investigated the potential for epigenetic regulation as a mechanism to explain the observed change in cellular oxidative capacity. However, we found no change to H3K79me3 signal in isolated histones after three days of 2HB treatment (*Figure 5—figure supplement 1C, D*). Neither C2C12s or MEFs exhibited any change to H3K27me3, H3K9me2, or H3K4me3, also suggesting no significant regulation of KDMs from three days of 2HB treatment (*Figure 5—figure supplement 1C, D*). Finally, we found no change to the expression levels of respiratory chain protein complexes I-V using an OXPHOS antibody cocktail (*Figure 5—figure supplement 1E, F*). These data do not support epigenetic regulation or mitochondrial biogenesis as mechanistic alternatives to SIRT4-dependent upregulation of BCAA degradation pathway for metabolic responses to chronic 2HB treatment.

## Repeated 2HB treatment replicates in vitro mechanisms

We next aimed to find evidence for mechanisms leading to the increased oxidative capacity in mice treated with 1 mmol/kg 2HB for one week. We focused analysis on the soleus muscle, as skeletal muscle accounts for the majority of BCAA degradation in the body and the oxidative muscle fibers found in soleus muscle possess greater mitochondrial content for BCAA degradation capacity (*Neinast et al., 2019*). We found that soleus muscles from mice treated with 2HB for 4 days exhibited increased ADPr in cytoplasmic extracts, but reduced ADPr in nuclear extracts (*Figure 6A*, *Figure 6—figure supplement 1A*). We repeated the poly/mono-ADPr and C/EBPβ immunoprecipitation experiments with soleus muscle nuclear extracts. We found that extracts from 2HB-treated mice exhibited reduced C/EBPβ ADPr (*Figure 6B*). Accordingly, we find the soleus from mice treated with 2HB for 1 week to exhibit increased gene expression of BCAT2, BCKDHA, and PCCA compared with controls (*Figure 6C*). We found increased protein expression of PCCA in 2HB-treated soleus lysates (*Figure 6D*). Further, we found increased BCKDHA protein expression and reduced the amount of BCKDHA phosphorylation at S293 in soleus samples, indicating increased flux through the BCAA degradation pathway (*Figure 6E*). No changes to BCKDHA expression or phosphorylation were observed in extensor digitorum longus (EDL) lysates (*Figure 6F*). These data support the mechanisms of 2HB-induced changes to metabolism found in vitro as being plausible in vivo and support the soleus as a site of response to exogenous 2HB.

As was observed in vitro, there was no change to protein expression of mitochondrial complexes I-V by western blot, indicating mitochondrial biogenesis as an unlikely explanation for the changes to oxidative capacity (*Figure 6—figure supplement 1B*). We also assessed histone methylation by

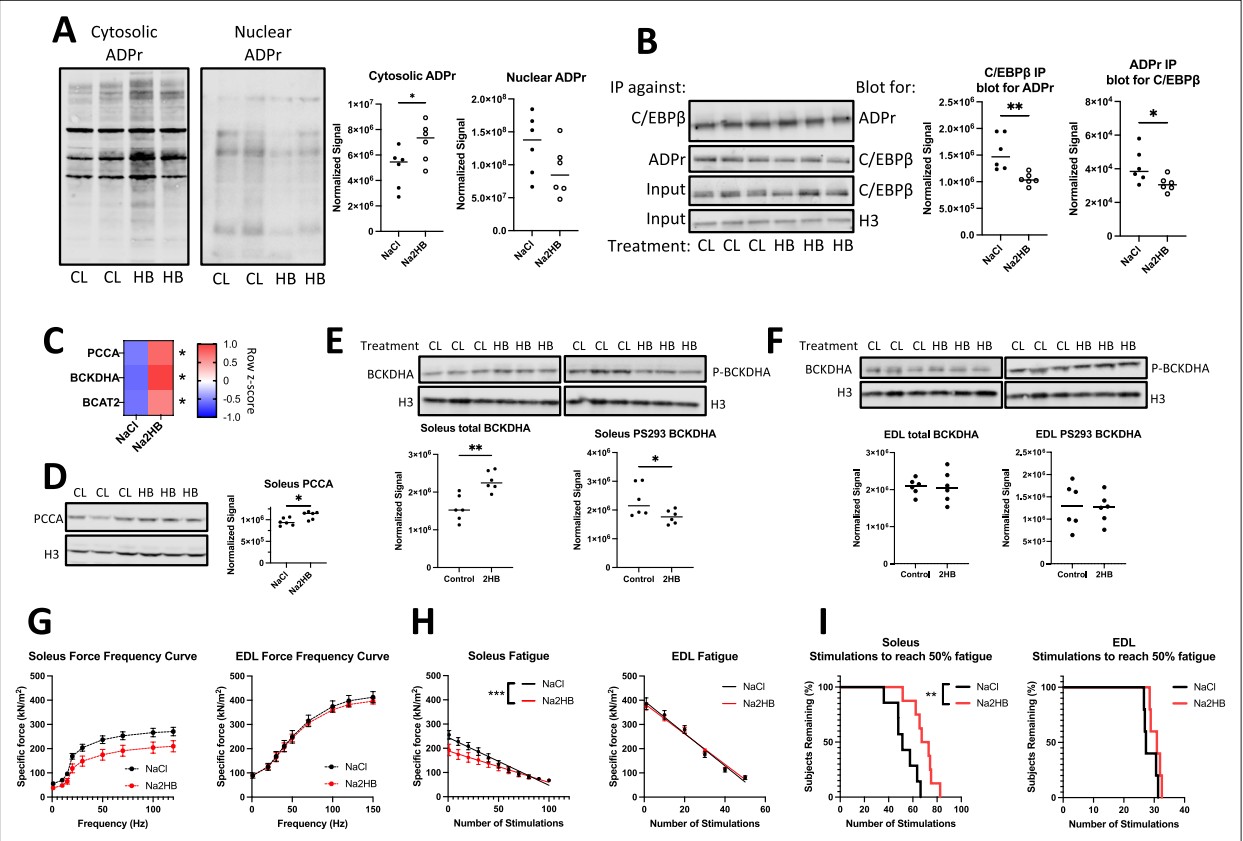

**Figure 6.** Mice treated with 2-hydroxybutyrate show functional responses intrinsic to skeletal muscle. Data come from mice treated for four days with 1 mmol/kg NaCl or Na2HB with harvest of soleus and EDL skeletal muscle on the fifth day. (**A**) Representative western blots and quantification for total protein ADP ribosylation from cytoplasmic and nuclear extracts of soleus muscle. (**B**) Immunoprecipitation was conducted on soleus nuclear extracts using either antibody against C/EBPβ or poly-ADP ribose, with resultant membranes probed using the opposite antibody. Total C/EBPβ and histone 3 were detected in input samples. For A,B, N=6, Student's t-test, * p0.05, ** p<0.01. (**C**) RT-qPCR of soleus muscle. N=4, Sidak multiple comparisons test * p<0.05. Representative western blots displayed using (**D,E**) soleus and (**F**) EDL lysates collected after incremental exercise tests showing total PCCA, BCKDHA, and BCKDHA phosphorylated at S293. N=10, student's t-test, * p<0.05. Ex vivo contractility analysis was conducted on freshly dissected muscle. (**G**) Force frequency curve, Sidak's multiple comparisons test found no differences. (**H**) Force production during fatigue protocol with linear regression line displayed. Slopes of linear regression lines are significantly different; F test *** p<0.001. (**I**) Time-to-event analysis displaying the number of stimulations in the fatigue protocol for the force output of each muscle to reach 50% of the initial force, log-rank test, ** p<0.01. N=7–8 for soleus, N=5 for EDL. For (**G, H**) mean ± SEM.

The online version of this article includes the following source data and figure supplement(s) for figure 6:

**Source data 1.** Raw data for *Figure 6*.

**Source data 2.** Labeled western blots.

**Source data 3.** Unlabeled western blots.

**Figure supplement 1.** Supplemental data regarding skeletal muscle response to 2-hydroxybutyrate treatment.

**Figure supplement 1—source data 1.** Raw data for *Figure 6—figure supplement 1*.

**Figure supplement 1—source data 2.** Labeled western blots.

**Figure supplement 1—source data 3.** Unlabeled western blots.

western blot and found no change to soleus H3K27me3, H3K79me3, or H3K4me3 after one week of treatment (*Figure 6—figure supplement 1C*).

## 2HB improves soleus muscle resistance to fatigue

We hypothesized that the improved performance on exercise tests provided by 2HB was dependent upon changes intrinsic to oxidative skeletal muscle, which exhibits the increased activation and expression of BCAA degradation factors. To investigate, we conducted ex vivo contractility analysis

of skeletal muscle isolated from mice treated with 1 mmol/kg/day Na2HB or NaCl control for four days. Each muscle was stimulated to produce a force-frequency curve, followed by a fatigue-inducing protocol. Soleus and EDL reached a peak specific force of approximately 200–270 kN/m² and 400 kN/m² respectively, with a trend towards reduced force production in 2HB-treated soleus (*Figure 6G*). Soleus muscles from 2HB treated mice demonstrated a slower rate of fatigue; the median number of stimulations required to reduce force production by 50% increased from 51 in controls to 70 in 2HB-treated soleus muscles (*Figure 6HI*). EDL from 2HB-treated mice performed similar to controls in the fatigue protocol (*Figure 6H,I*). These data demonstrate a functional change intrinsic to the skeletal muscle of 2HB-treated mice, specific to oxidative muscle groups.

## Discussion

2HB has to date been described as a waste product that is released into the circulation during periods of metabolic or oxidative stress, for example, in acute infection, or post-exercise (*Morville et al., 2020*; *Contrepois et al., 2020*; *Bruzzone et al., 2020*). Here, we describe an axis of cellular responses to increased levels of 2HB which regulate the BCAA degradation pathway. We find that exogenous 2HB treatment can recapitulate the effects of short-term exercise training in mice, via improving the fatigue resistance of oxidative skeletal muscle. These data provide insight into novel mechanisms involved in physiological changes to metabolism during exercise training.

The accumulation of 2HB is dependent upon the metabolic balance of 2 KB. We demonstrate that 2 KB is a suitable fuel for mitochondrial oxidation, dependent upon an intact BCAA degradation pathway (*Figure 3*). We found that PCC was required for 2 KB oxidation, while the BCKDH complex was dispensable. This is in line with data showing that pyruvate dehydrogenase may also oxidize 2 KB, although this too has been shown to be dispensable (*Thompson Legault et al., 2015*; *Steele et al., 1984*; *Paxton et al., 1986*). As the capacity for 2 KB oxidation will modify 2HB production, future work interested in the presented 2HB-dependent mechanism should carefully consider the expression of pyruvate dehydrogenase and BCKDH in the tissue of interest as well as the metabolic environment guiding flux through these enzymes. Alternatively, 2 KB may be reduced by LDH to 2HB. The balance of oxidation versus reduction likely depends on cellular demand for NADH, oxygen availability, and capacity within the BCAA degradation pathway (*Anderson et al., 2017*; *Zaganjor et al., 2021*; *Min et al., 2018*; *White et al., 2018*).

We find that 2HB is a poor substrate for LDH oxidation, supporting the well-described accumulation of 2HB post-exercise, with a slow return to homeostatic levels (*Figures 2 and 3*; *Sato et al., 2022*; *Morville et al., 2020*). Together, these observations are consistent with the accumulation of 2HB as an indicator of saturated capacity for 2 KB oxidation via the BCAA degradation pathway. Interestingly, this pattern is mirrored by the BCHA, 2-hydroxyisocaproate, 2-hydroxy-3-methylvalerate, and 2-hydroxyisovalerate in human plasma metabolomics following exercise (*Figure 2*). The BCHA are rare metabolites, which is consistent with our observation of very low relative reaction rates for the reduction of BCKA by LDH (*Figure 3—figure supplement 1*). Further, we were unable to observe significant oxidation of the BCHA in isolated LDH assays (*Figure 3—figure supplement 1*), which is consistent with the pattern of accumulation without return to baseline for all the hydroxy acids (*Figure 2*). Thus, we propose that the reduction of 2 KB and BCKA indicate oversaturation of the BCAA degradation pathway, leading to the LDH-dependent production of 2HB or BCHA that are metabolically stubborn and tend to accumulate.

It is reported that the liver is the major producer of circulating 2HB (*Sato et al., 2022*), which would suggest that many tissues receive external 2HB, and that therefore the balance of 2HB metabolism will depend upon 2HB oxidation. LDH isolated from bovine heart had a greater capacity for oxidation of 2HB than LDH isolated from bovine skeletal muscle (*Figure 3*), suggesting that *LDHB* gene expression may provide an advantage when metabolising exogenous 2HB. Skeletal muscle expression of *LDHB* was previously reported to be induced by exercise training, and to be dependent upon peroxisome proliferator-activated receptor γ coactivator 1α (PGC-1α), and that *LDHB* expression was required for generating an exercise training benefit (*Liang et al., 2016*). Together, this would suggest that improved capacity for metabolising 2HB is common to the progression of endurance training. Since we observed no induction to PGC-1α gene expression or mitochondrial biogenesis (*Figure 5—figure supplement 1*, *Figure 6—figure supplement 1*), treatment with exogenous 2HB does not appear to replicate all exercise training responses in oxidative skeletal muscle. It is possible that 2HB-induced

effects on BCAA metabolism occur in an early stage of skeletal muscle endurance training, and that additional signals are required to continue training progression.

We found that the in vitro improvement to oxidative capacity induced by 2HB was dependent upon expression of BCKDHA and SIRT4 (*Figure 5*). SIRT4 has been previously reported to produce greater expression and activation of BCAA degradation pathway members (*Anderson et al., 2017*; *Zaganjor et al., 2021*), along with C/EBPβ (*Ryu et al., 2018*), making this a logical signaling axis for 2HB response. SIRT4 is relatively unique in its preference for ADP-ribosylation activity as opposed to the de-acylation activity of other sirtuins (*Kumar and Lombard, 2015*), providing further support for the role of SIRT4 in the presented mechanism. However, future work should investigate the role of other sirtuins, particularly the other mitochondrial sirtuins SIRT3 and SIRT5, as these are often more prevalent than SIRT4 and modify a large range of target proteins.

Increased flux in the BCAA degradation pathway is reflected by a reduced proportion of BCKDHA phosphorylation (*White et al., 2018*; *Paxton and Harris, 1984*), which we observe in the soleus of 2HB-treated mice post-exercise (*Figure 6*). BCKDHA phosphorylation by BCKDH kinase is inhibited by BCKA. Thus, the reduced BCKDHA phosphorylation is consistent with the increase to BCKA observed in oxidative red gastrocnemius muscle (*Figure 2*, *Figure 2—figure supplement 1*). That we observe reduced levels of muscular BCHA despite the increase to BCKA levels, without an increase in release of BCKA to the serum, suggests an increase to the oxidative muscle capacity for BCKA oxidation (*Figure 2*).

Together, these data form a logical feedback loop where accumulation of 2HB provides negative feedback to reduce BCAA degradation at the stage of BCAT. Acutely, this would reduce competition for 2 KB in the BCAA degradation pathway by slowing conversion of BCAA to keto acids. A sufficiently large acute dose of 2HB may limit mitochondrial fuel by inhibiting BCAT. This model is consistent with the observed rapid increase to NAD+/NADH ratio in MEFs and C2C12s treated with 2HB in culture (*Figure 4*). We show that this metabolic effect will lead to transcriptional feedback; an increase BCAA and 2 KB oxidation capacity will in the long-term reduce the production of 2HB and BCHA, as observed in red gastrocnemius post-exercise in 2HB-treated mice (*Figure 2*, *Figure 2—figure supplement 1*). That we observe similar transcriptional regulation, changes to BCKDHA phosphorylation, and changes to oxidative metabolism phenotypes in vivo suggests that a similar mechanism is occurring in response to intraperitoneal injections of 2HB. In the case of exercise, the physiological accumulation of 2HB would occur during high metabolic demand when mitochondrial fuel supply may be sensitive to the 2HB despite a potentially lower concentration than the exogenous doses used in this study. This is supported by the lack of additive effect on mouse $\Delta VO_2max$ when exogenous 2HB treatment was combined with exercise training (*Figure 1*).

Overall, this study presents an axis of cellular response to 2HB and presents a functional response to repeated 2HB treatment in the form of improved exercise performance and resistance to fatigue in oxidative murine skeletal muscle. Further work investigating cellular responses to accumulated 2HB are warranted as this metabolite accumulates in the circulation of individuals with infection and metabolic disorders (*Bruzzone et al., 2020*; *Gall et al., 2010*; *Cobb et al., 2016*).

## Materials and methods
### Animals
Male C57Bl/6 mice were purchased from Janvier Labs (C57Bl/6JRj) and housed at Karolinska Institutet under specific pathogen-free conditions. All experiments and protocols were approved by the regional ethics committee of Northern Stockholm (Ref. 19683–2021). Six-week-old mice were housed for 2 weeks prior to any experimental procedures to acclimate to the new housing. Mice were randomly assigned to treatment groups following acclimation. At experimental endpoint mice were terminated by carbon dioxide. Blood was collected via cardiac puncture and spun in tabletop centrifuge to remove coagulated components, yielding serum. Collected tissue was snap-frozen in liquid nitrogen.

## Incremental exercise protocols

An enclosed chamber treadmill (Columbus Instruments) set at 10° incline was used for all exercise and respiration measurements. Eight-week-old mice were acclimated to the treadmill progressively over 4 days in a standardized protocol across all mice:

Day 1: mice are placed on stationary treadmill for 5 min. Starting at 6 m/min for 3 min, speed is increased 3 m/min every 3 minutes. Completed after 3 min at 12 m/min.

Day 2: mice are placed on stationary treadmill for 3 min. Starting at 6 m/min for 3 min, speed is increased 3 m/min every 3 min. Completed after 3 min at 15 m/min.

Day 3: mice are placed on stationary treadmill for 3 min. Starting at 9 m/min for 3 min, speed is increased 3 m/min every 3 minutes. Completed after 3 min at 18 m/min.

Day 4: mice are placed on stationary treadmill for 3 min. Starting at 9 m/min for 3 min, speed is increased 3 m/min every 3 min. Completed after 3 min at 21 m/min.

Acclimation was completed at least 48 hr prior to any incremental exercise tests concurrent with measurements of $O_2$ and $CO_2$ respiration. Incremental exercise tests began with a 5-min settling period, 3 min of measurement with the treadmill at rest, followed by an initial treadmill speed of 6 m/min, after which the speed was increased by 3 m/min every 3 min. Exercise and measurements of oxygen consumption continued until mouse exhaustion, defined as when the mouse refused to run after 10 continuous seconds in contact with an electrical grid providing a 1.22 mAmp stimulus at a rate of 2 Hz. *Figure 2* exercise held the treadmill at 18 m/min for 30 min instead of continuing until exhaustion. Oxygen and carbon dioxide was monitored using a carbon dioxide and paramagnetic oxygen sensor within an OxyMax system (Columbus Instruments). Measurements were collected every 15 s. Basal $VO_2$ measured while the treadmill was at rest was used to calculate the change to $VO_2$ ($\Delta VO_2$) throughout the exercise protocol. $\Delta VO_2$max was defined as the maximum increase to $VO_2$ above baseline achieved during the incremental exercise protocol.

## Cells

Mouse embryonic fibroblasts (MEFs) were originally derived as described previously (*Sim et al., 2018*). C2C12 murine myoblasts were a gift from Dr. Jorge Ruas. Cells were cultured in Dulbecco's modified eagle medium (DMEM) supplemented with 10% fetal bovine serum and penicillin/streptomycin. Cells were cultured with 5% $CO_2$, humidity and temperature control, in a cell culture incubator in normoxia (21% $O_2$, Sanyo).

## Compounds and drugs

Cell culture media was supplemented as indicated with sodium salts of metabolites, including Na2-hydroxybutyrate and Na2-ketobutyrate and NaCl as a salinity control. Na2HB and NaCl were dissolved in sterile PBS and administered to mice at a concentration of 250 mM with a dosage of 1 mmol/kg. Aminooxyacetic acid (AOA; Sigma) was used as an inhibitor of aminotransferases, and the DOT1L inhibitor EPZ004777 (MedChemExpress).

## Expression plasmids and transfections

MEF and C2C12 cells were transiently transfected using Lipofectamine 2000 (Thermo Fisher) with 20 ng per expression plasmid for $2x10^4$ cells per well in a 96-well plate. Cells were treated with transfection mixes for 8–24 hr before further treatment. Knockdown experiments used dicer-substrate siRNA (IDT DNA) against BCKDHA, PCCA, or SIRT4, sequences in Supplementary Information. Knockdown was confirmed by RT-qPCR 24 hr after transfection.

## In vitro oxygen consumption rate and extracellular acidification rate

Cellular oxygen consumption rate (OCR) was measured using a Seahorse Extracellular Flux Analyser XF96 (Agilent). MEFs or C2C12s were seeded with $2x10^4$ cells per well on the day prior to assay. XF DMEM media (Agilent) was supplemented with 10 mM glucose, 2 mM of L-glutamine, and 1 mM Na pyruvate as indicated. Cells were subjected to a 'mitochondrial stress test' protocol, involving sequential injection of 2.5 µM oligomycin, 1 µM FCCP, and a combination of 1 µM antimycin A with 100 nM rotenone. Cells were also subjected to a standard 'glycolysis stress test' protocol with quantification of the extracellular acidification rate (ECAR), involving sequential injection of 10 mM glucose, 2.5 µM oligomycin, and 50 mM 2-deoxyglucose. Listed concentrations are the final concentrations for each

compound. All compounds added in seahorse assays were acquired from Sigma. For experiments acutely treating cells with metabolites within the seahorse instrument, OCR and ECAR values were normalised to baseline measures. For experiments investigating the effects of treatments prior to the assay, OCR and ECAR values were normalized to protein levels within each well quantified by BCA assay (Abcam) conducted after the Seahorse assay.

## Western blotting

Total protein was isolated from cells or mouse tissue using RIPA buffer (Thermo Fisher), or lysates were enriched for histone proteins using Histone Extraction Kit (Abcam), or cytoplasmic and nuclear extracts were isolated using NE-PER Nuclear and Cytoplasmic Extraction Reagents (ThermoFisher), each according to manufacturer instructions. Tissues were homogenized using a glass dounce homogenizer (Active Motif) and kept on ice during homogenization. Proteins were separated in SDS-PAGE, transferred onto PVDF membrane, and probed with antibodies against C/EBPβ (Thermo Fisher, MA1-827), BCKDHA (Thermo Fisher, PA597248), BCAT2 (abcam, ab95976), DDK tag (Origene, TA50011-100), Histone 3 (abcam, ab12079), H3K4me3 (Cell Signaling Technologies, 62255), H3K9me2 (Cell Signaling Technologies, 4658), H3K27me3 (Cell Signaling Technologies, 9733), H3K79me3 (Cell Signaling Technologies, 4260), OXPHOS antibody cocktail (abcam, ab110413), or Poly/Mono ADP-ribose (Cell Signaling Technologies, 83732) and detected using infrared labeled secondary antibody and an ODYSSEY imaging system (LICOR).Antibody signal was normalized to total protein quantified by REVERT total protein stain (LICOR) or appropriate loading control. Western blot data were analyzed using Image Studio Lite software (Image Studio Lite).

## RT-qPCR

Total RNA was extracted using Trizol (Thermo Fisher) according to manufacturer instructions. One microgram of RNA was reverse transcribed using iScript cDNA synthesis kit (BioRad) in a total volume of 20 µL. Real time RT-RT-qPCR was applied to quantify mRNA (7500 Fast Real-Time PCR system, Applied Biosciences Inc, or StepOnePlus Real-time PCR system, Thermo Fisher). Primers are listed in Supplementary Information. Reactions were performed in 96-well MicroAmp Optical plates in duplicate, using SsoAdvanced Universal SYBR Green supermix (BioRad). Gene expression was normalized to *HPRT* unless otherwise indicated.

## Metabolomics analysis

Metabolomics data from *Sato et al., 2022* and *Rundqvist et al., 2020* was selected based on relevance to our hypotheses. Data was scaled before correlation analysis and partial correlation coefficient calculated to quantify metabolite pair correlations. Correlation analysis was conducted using the MetScape correlation calculator (*Karnovsky et al., 2012*).

Metabolomics related to *Figure 2* was performed by Metabolon. Peak area data was subjected to feature-wise normalization using standard scaling or log-transformation to optimize data column normality (*Pedregosa et al., 2011*). Normalization and scaling were conducted separately for serum, red gastrocnemius, and white gastrocnemius data. Exercise and sham treatments for metabolomics experiments occurred inside of sealed treadmills with respiration measurements taken throughout exercise. As treatment altered exercise performance, for some analyses metabolite data was adjusted based on multiple linear regression of respiration parameters. Metabolomics data analysis was conducted using custom python code. Statistical comparisons conducted using statsmodels python package (*Seabold and Perktold, 2010*). Heatmaps produced using seaborn python package (*Waskom, 2021*) or Prism9 (GraphPad).

## LDH, BCAT, KDM6A, Prolyl hydroxylase activity assays

Activity of LDH was assessed using of isolated bovine heart LDH (Sigma), isolated bovine muscle LDH (Sigma), or mouse tissue lysates as described previously (*Talaiezadeh et al., 2015*). Briefly, one unit of isolated LDH enzyme or 5 µg of tissue lysate was added to a solution of substrate with 2 mM NAD +for oxidation reactions or substrate with 0.5 mM NADH for reduction reactions, in 20 mM Tris-HCl buffer pH 8.0. Activity of BCAT was assessed using recombinant human BCAT2 (rhBCAT2; R&D Systems; *Cooper et al., 2002*). Briefly, the reaction catalyzed by BCAT2 transferring an amino group from leucine (Leu) to αKG, producing ketoisocaproate (KIC) and glutamate (Glu) respectively, is

coupled to the NADH-consuming reaction catalyzed by leucine dehydrogenase from *Bacillus cereus* (LeuDH; Sigma), which converts KIC back to Leu. The reaction mixture consisted of 5 μM pyridoxal phosphate, 50 mM ammonium sulfate, 5 mM DTT, a range of αKG, 10 mM Leu, 0.5 mM NADH, and a range of Na2HB in 100 mM potassium phosphate buffer pH 7.4. One unit of LeuDH along with 100 ng of rhBCAT2 was added to the reaction mixture to initiate the reaction. Upon addition of enzyme, reaction plates were immediately transferred to a plate reader (Synergy Biotek) set at 37 °C and absorbance at 340 nm quantified every 30 s for 10 min. The initial maximum rate of reaction following any present lag phase was abstracted as the reaction rate.

Activity assays for human KDM6A and the prolyl hydroxylases, HIFP4H1 and HIFP4H2, were conducted as described previously (*Chakraborty et al., 2019*; *Hirsilä et al., 2003*; *Minogue et al., 2023*).

## Skeletal muscle ex vivo contractile function and fatigue

Skeletal muscle was prepared and assessed as described previously (*Chaillou et al., 2017*). Briefly, the soleus and EDL muscle were excised under dissection microscope from the right hindlimb with proximal and distal tendons kept intact. Excess adipose tissue was manually cleaned from the muscles and tendons were tied with a braided silk thread and mounted in a 15 mL stimulation chamber between a force transducer and an adjustable holder (World Precision Instruments). Muscles were submerged in a Tyrode solution containing (in mmol/L): 121 NaCl, 5 KCl, 1.8 $CaCl_2$, 0.1 EDTA, 0.5 $MgCl_2$, 0.4 $NaH_2PO_4$, 5.5 glucose, and 24 $NaHCO_3$. The force stimulation chamber was set to 31 °C and gassed with 95% $O_2$ 5% $CO_2$ for a pH of 7.4. Each mounted muscle was adjusted to the length at which the highest twitch force was recorded and then allowed to rest for 15 min. The force frequency relationship was determined through stimulations at the following frequencies: 1 (twitch), 10, 15, 20, 30, 50, 70, 100, and 120 Hz for soleus, and 1, 20, 30, 40, 50, 70, 100, 120, and 150 Hz for EDL. The stimuli were interspaced by 1 min of rest. Upon completion, the muscle was allowed to rest for 7 min before starting a fatigue protocol consisting of 100 stimulations (70 Hz, 600 ms train duration, 2 s interval duration) for the soleus or 50 stimulation (100 Hz, 300 ms train duration, 2 s interval duration) for the EDL. Muscle recovery was determined at 1, 2, 5, and 10 min after the final tetanic stimulation at the same stimulation frequency. Absolute force was expressed in millinewton (mN). Specific force ($kN/m^2$) was calculated by dividing the absolute force by the muscle cross sectional area, the latter determined by dividing the muscle mass (with tendons removed) by muscle length and density, assuming a density of 1.06 $g/cm^3$.

## Statistical analysis

Data was visualized and statistical analyses conducted in Prism 9 (GraphPad). Statistical tests and replicates are stated in figure legends. Plotted data represent biological replicates of either distinct animals for in vivo and ex vivo experiments, or independent assays for in vitro experiments.

## Acknowledgements

We acknowledge Dr. Cristina M. Branco of Queen's University, Belfast, for her role and support in the supervision and funding of Pedro P. Cunha.

## Additional information

### Funding

| Funder | Grant reference number | Author |
|---|---|---|
| Vetenskapsrådet | 2022-00755 | Randall S Johnson |
| Barncancerfonden | PR2023-0072 | Randall S Johnson |
| Wellcome Trust | 10.35802/214283 | Randall S Johnson |
| Canadian Institutes of Health Research | | Brennan J Wadsworth |

| Funder | Grant reference number | Author |
| --- | --- | --- |
| Cancerfonden | 180676 | Randall S Johnson |
| Knut och Alice Wallenbergs Stiftelse | 2023.0350 | Randall S Johnson |
| Fundação para a Ciência e a Tecnologia | SFRH/BD/115612/2016 | Pedro P Cunha |

The funders had no role in study design, data collection and interpretation, or the decision to submit the work for publication. For the purpose of Open Access, the authors have applied a CC BY public copyright license to any Author Accepted Manuscript version arising from this submission.

## Author contributions

Brennan J Wadsworth, Conceptualization, Data curation, Formal analysis, Supervision, Funding acquisition, Validation, Investigation, Visualization, Methodology, Writing - original draft, Writing – review and editing; Marina Leiwe, Pedro P Cunha, Helene Rundqvist, Formal analysis, Investigation, Methodology, Writing – review and editing; Eleanor A Minogue, Viktor Engman, Carolin Brombach, Christos Asvestis, Shiv K Sah-Teli, Peppi Koivunen, Investigation, Methodology, Writing – review and editing; Emilia Marklund, Investigation; Jorge L Ruas, Investigation, Writing – review and editing; Johanna T Lanner, Formal analysis, Supervision, Funding acquisition, Methodology, Writing – review and editing; Randall S Johnson, Conceptualization, Resources, Formal analysis, Funding acquisition, Project administration, Writing – review and editing

## Author ORCIDs

Brennan J Wadsworth https://orcid.org/0000-0002-9183-227X
Marina Leiwe https://orcid.org/0009-0006-7936-6356
Pedro P Cunha https://orcid.org/0000-0002-3814-6289
Christos Asvestis https://orcid.org/0009-0003-5565-030X
Peppi Koivunen https://orcid.org/0000-0002-2827-8229
Jorge L Ruas https://orcid.org/0000-0002-1110-2606
Helene Rundqvist https://orcid.org/0000-0002-5617-9076
Randall S Johnson https://orcid.org/0000-0002-4084-6639

## Ethics

This study was performed in accordance with the recommendations for ethical experimentation of Swedish and EU laws and regulations. All animal handling was performed in accordance with approved institutional animal care protocols. Animal protocol was approved by the Committee on Ethics of Animal Experiments of the Ethical Review Board of Northern Stockholm (Ref. 19683-2021) and the Swedish Ministry of Agriculture.

Reviewer #1 (Public Review): https://doi.org/10.7554/eLife.92707.2.sa1
Reviewer #2 (Public Review): https://doi.org/10.7554/eLife.92707.2.sa2
Author response https://doi.org/10.7554/eLife.92707.2.sa3

# Additional files

## Supplementary files
• Supplementary file 1. 2HB and 2KB IC50 for isolated PHD and KDM enzymes.
• MDAR checklist

## Data availability

All data generated or analysed and plotted to produce this manuscript are included in the source data file(s). Original code used to analyse metabolomics data is available at https://github.com/bjwadswo/JohnsonLab_Wadsworth-et-al_2HB-Metabolomics (copy archived at *Wadsworth, 2024*).

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
