## [Editor Report · eLife assessment]

The work by Johnson and co-workers has identified an **important** role of 2-Hydroxybutyrate in skeletal muscle oxidative capacity in the early stages of exercise. Mechanistically, they show **convincing** data to support a role of 2-Hydroxybutyrate in the regulation of BCAA metabolism via SIRT4, ADP-Ribosylation, and CEBP. However, whether this is the sole mechanism and if these translate to longer exercise training regimes requires future experiments.

---

## [Referee Report · Reviewer #1 (Public Review)]

The authors aimed to investigate if 2-hydroxybutyrate (2HB), a metabolite induced by exercise, influences physiological changes, particularly metabolic alterations post-exercise training. They treated young mice and cultured myoblasts with 2HB, conducted exercise tests, metabolomic profiling, gene expression analysis, and knockdown experiments to understand 2HB's mechanisms. Their findings indicate that 2HB enhances exercise tolerance, boosts branch chain amino acid (BCAA) enzyme gene expression in skeletal muscles, and increases oxidative capacity. They also highlight the role of SIRT4 in these effects. This study establishes 2HB, once considered a waste product, as a regulator of exercise-induced metabolic processes. The study's strength lies in its consistent results across in vitro, in vivo, and ex vivo analyses. The authors propose a mechanism in which 2HB inhibits BCAA breakdown, raises NAD+/NADH ratio, activates SIRT4, increases ADP ribosylation, and controls gene expression.

However, some questions remain unclear based on these findings:

This study focused on the effects of short-term exercise (1 or 5 bouts of treadmill running) and short-term 2HB treatment (1 or 4 days of treatment). Adaptations to exercise training typically occur progressively over an extended period. It's important to investigate the effects of long-term 2HB treatment and whether extended combined 2HB treatment and exercise training have independent, synergistic, or antagonistic effects.

Exercise training leads to significant mitochondrial changes, including increased mitochondrial biogenesis in skeletal muscle. It would be valuable to compare the impact of 2HB treatment on mitochondrial content and oxidative capacity in treated mice to that in exercised mice.

The authors demonstrate that 2-ketobutyrate (2KB) can serve as an oxidative fuel, suggesting a role for the intact BCAA catabolic pathway. However, it's puzzling that the knockout of BCKDHA, a subunit crucial for the second step of BCAA catabolism, did not result in changes in oxidative capacity in cultured myoblasts.

Nevertheless, this innovative model of metabolic signaling during exercise will serve as a valuable reference for informing future.

---

## [Referee Report · Reviewer #2 (Public Review)]

Summary:

The manuscript entitled "A 2-HB-mediated feedback loop regulates muscular fatigue" by the Johnson group reports interesting findings with implications for the health benefits of exercise. The authors use a combination of metabolic/biochemical in vivo and in vitro assays to delineate a metabolic route triggered by 2-HB (a relatively stable metabolite induced by exercise in humans and mice) that controls branched-chain amino transferase enzymes and mitochondrial oxidative capacity. Mechanistically, the author shows that 2-HB is a direct inhibitor of BCAT enzymes that in turn control levels of SIRT4 activity and ADp-ribosylation in the nucleus targeting C/EBP transcription factor, affecting BCAA oxidation genes (see Fig 4i in the paper). Overall, these are interesting and novel observations and findings with relevance to human exercise, with the potential implication of using these metabolites to mimic exercise benefits, or conditions or muscular fatigue that occurs in different human chronic diseases including rheumatic diseases or long COVID.

Weaknesses:

There are several experiments/comments that will strengthen the manuscript-

1- A final model in Figure 6 integrating the exercise/mechanistic findings, expanding on Fig 4i will clarify the findings.

2- In some of the graphs, statistics are missing (e.g Fig 6G).

3- The conclusions on SIRT4 dependency should be carefully written, as it is likely that this is only one potential mechanism, further validation with mouse models would be necessary.

4- One of the needed experiments to support the oxidative capacity effects that could be done in cultured cells, is the use of radiosotope metabolites including BCCAs to determine the ability to produce CO2. Alternatively or in combination metabolite flux using isotopes would be useful to strengthen the current results.

---

## [Author Response]

**Reviewer #1 (Public Review):**
The authors aimed to investigate if 2-hydroxybutyrate (2HB), a metabolite induced by exercise, influences physiological changes, particularly metabolic alterations post-exercise training. They treated young mice and cultured myoblasts with 2HB, conducted exercise tests, metabolomic profiling, gene expression analysis, and knockdown experiments to understand 2HB's mechanisms. Their findings indicate that 2HB enhances exercise tolerance, boosts branch chain amino acid (BCAA) enzyme gene expression in skeletal muscles, and increases oxidative capacity. They also highlight the role of SIRT4 in these effects. This study establishes 2HB, once considered a waste product, as a regulator of exercise-induced metabolic processes. The study's strength lies in its consistent results across in vitro, in vivo, and ex vivo analyses.The authors propose a mechanism in which 2HB inhibits BCAA breakdown, raises NAD+/NADH ratio, activates SIRT4, increases ADP ribosylation, and controls gene expression.However, some questions remain unclear based on these findings:This study focused on the effects of short-term exercise (1 or 5 bouts of treadmill running) and short-term 2HB treatment (1 or 4 days of treatment). Adaptations to exercise training typically occur progressively over an extended period. It's important to investigate the effects of long-term 2HB treatment and whether extended combined 2HB treatment and exercise training have independent, synergistic, or antagonistic effects.

We agree with the reviewer that investigation of longer-term 2HB treatment may potentially yield interesting findings with more implications to exercise physiology. To investigate the effects of 2HB treatment against or in combination with a progressive exercise training protocol would require an experiment duration between 4 to 12 weeks, based on previous studies (Systematic Review by Massett et al., Frontiers in Physiology, 2021, 10.3389/fphys.2021.782695). However, our experience with these types of experiments is that such a pursuit would require a breadth of work beyond the scope of this current study. For instance, if there were evidence of weakened effect of 2HB over time, one may be compelled to investigate other organs such as the liver to find signs of metabolic adaptation to the exogenous metabolite. If there were additive or synergistic effects on exercise performance, one may be compelled to investigate changes to the cardiovascular system in addition to the skeletal muscle. Additional questions would be raised around the skeletal muscle as well, including assessment of structural and fibre-type changes. Further, these additional mechanisms would need to be characterized in a time course fashion. Rather, we view the scope of the current study to be the acute response to 2HB as an initial report on mechanistic effects of 2HB.

Exercise training leads to significant mitochondrial changes, including increased mitochondrial biogenesis in skeletal muscle. It would be valuable to compare the impact of 2HB treatment on mitochondrial content and oxidative capacity in treated mice to that in exercised mice.

We agree with the author that it is of interest to investigate how 2HB may affect mitochondrial biogenesis. However, our preliminary findings were that 2HB-treated MEFs, C2C12s, and mouse soleus muscles showed no change in PGC1α gene expression after four days of treatment (data not shown). As a follow-up assessment of mitochondrial protein expression, although not specific to mtDNA derived genes, we quantified the expression of the respiratory chain proteins in cells and soleus muscle and found no effect of 2HB treatment (SFig. 5,6). At this stage we conclude that there is not evidence of 2HB modifying mitochondrial biogenesis in this time frame and that further investigation would be best suited to a follow-up study such as one interested in long-term exercise training.

The authors demonstrate that 2-ketobutyrate (2KB) can serve as an oxidative fuel, suggesting a role for the intact BCAA catabolic pathway. However, it's puzzling that the knockout of BCKDHA, a subunit crucial for the second step of BCAA catabolism, did not result in changes in oxidative capacity in cultured myoblasts.

While we report the BCKDH complex to be dispensable for 2KB oxidation it is important to note that previous studies have reported the following: (1) that 2KB is a viable substrate for BCKDH, (2) that 2KB is a viable substrate for pyruvate dehydrogenase, and (3) that pyruvate dehydrogenase is also dispensable for 2KB oxidation (see Steele et al., J Nutr., 114: 701-710, and Paxton et al. Biochem J., 234:295-303). Collectively, these data have led previous studies to conclude that BCKDH and pyruvate dehydrogenase are redundant for the first step of 2KB oxidation, with a preference for BCKDH. The flux through either may depend upon the metabolic environment. The aim for figure 3C was to determine whether the BCAA degradation pathway was required for 2KB oxidation. We conclude that this pathway is required, first at the step of PCC.

While these past studies were mentioned in paragraph 2 of the discussion, in light of thereviewer’s comment we have expanded this paragraph. We have added language to explain that future research interested in the presented 2HB mechanism should carefully consider BCKDH and PDH expression in the cell or tissue of interest, as the metabolism of 2KB is quite central to the presented mechanism.

Nevertheless, this innovative model of metabolic signaling during exercise will serve as a valuable reference for informing future.
**Reviewer #2 (Public Review):**
Summary:The manuscript entitled "A 2-HB-mediated feedback loop regulates muscular fatigue" by the Johnson group reports interesting findings with implications for the health benefits of exercise. The authors use a combination of metabolic/biochemical in vivo and in vitro assays to delineate a metabolic route triggered by 2-HB (a relatively stable metabolite induced by exercise in humans and mice) that controls branched-chain amino transferase enzymes and mitochondrial oxidative capacity. Mechanistically, the author shows that 2-HB is a direct inhibitor of BCAT enzymes that in turn control levels of SIRT4 activity and ADp-ribosylation in the nucleus targeting C/EBP transcription factor, affecting BCAA oxidation genes (see Fig 4i in the paper). Overall, these are interesting and novel observations and findings with relevance to human exercise, with the potential implication of using these metabolites to mimic exercise benefits, or conditions or muscular fatigue that occurs in different human chronic diseases including rheumatic diseases or long COVID.Weaknesses:There are several experiments/comments that will strengthen the manuscript-(1) A final model in Figure 6 integrating the exercise/mechanistic findings, expanding on Fig 4i will clarify the findings.

We appreciate the reviewer’s suggestion to incorporate the exercise findings into a summary figure. However, upon internal review we find that such a figure is too similar to Fig 4i to warrant a new diagram.

(2) In some of the graphs, statistics are missing (e.g Fig 6G).

Some figures are included primarily for the reader to visualize the data while statistical comparison is conducted in a separate figure, for example Fig 2D-G. However, we have revised the figure legends to ensure that statistical comparisons are described for all appropriate figures, including Fig 6G identified by the reviewer.

(3) The conclusions on SIRT4 dependency should be carefully written, as it is likely that this is only one potential mechanism, further validation with mouse models would be necessary.

We appreciate the reviewers feedback and take the point well that a NAD-dependent mechanism will likely stimulate other sirtuins, which are often in fact expressed at greater levels than SIRT4. To reflect this comment in the manuscript we have altered paragraph 5 of the discussion to now focus on sirtuins. We briefly discuss SIRT4 and highlight the need for future consideration of other sirtuins, perhaps particularly mitochondrial sirtuins.

(4) One of the needed experiments to support the oxidative capacity effects that could be done in cultured cells, is the use of radiosotope metabolites including BCCAs to determine the ability to produce CO2. Alternatively or in combination metabolite flux using isotopes would be useful to strengthen the current results.

We appreciate the suggestion from the reviewer and we will look to conduct such an experiment in our follow-up work.

We sincerely thank the reviewers for their input on this study as their suggestions have led to an improved manuscript for the version of record. The reviewer comments are well taken and we are glad that they will be present alongside the final manuscript to provide an important perspective on the work.